# Research-informed decision-making for empowering integrated care system development: Co-creating innovative solutions to facilitate enhanced service provision

**Eren Demir**[ID]* **Usame Yakutcan, Stephen Page**

Hertfordshire Business School, University of Hertfordshire,, Hatfield, United Kingdom

* e.demir@herts.ac.uk

## Abstract

Integrated care has emerged as a vital approach to addressing complex health and social care challenges through attempting to foster collaborative provision in healthcare settings. Yet as demand for services often outstrips supply, hospitals, as anchor institutions in communities, are constantly seeking to innovate to align their resources with needs and policy priorities. As hospitals are often viewed as a conduit for creating and embracing innovation to enhance organisational performance, this paper outlines one such innovation, which was co-created as a partnership between a university and hospital to help it with its transition to an integrated care system (ICS). By developing a full hospital system model in partnership not only with hospital stakeholders but also out-of-hospital services - such as community and primary care - for an integrated care model, this study helps to translate an innovative model into practice at an ICS level. To achieve this, decision support tools (DSTs) were used to foster evidence-based assessment of the hospital system, and key opinion leaders (KOLs) were provided with a versatile toolset with which to optimise workforce productivity and deployment, innovate service provision, and enhance community health.

## Introduction

At a global scale, various national governments have pursued different models of healthcare to meet a myriad of human, financial, and management challenges (e.g., ageing populations, an increasing burden of chronic diseases, workforce shortages, healthcare inequalities, and rising costs of care) [1]. Many of these models have been determined by prevailing political ideology (i.e., the extent to which either state or private funding is preferred by policymakers). Toth [2] outlined the variety of models of healthcare that existed in 20 OECD countries; these ranged from separated models of healthcare (i.e., with significant elements of private insurance), via moderately separated or moderately integrated systems through to fully integrated models of care, typically funded by the state. In a study of non-industrialised countries, Reid [3] also outlined the prevailing out-of-pocket expense model, in which the state is unable to provide

**Data availability statement:** All relevant data are within the manuscript and its supporting information files (see Table 3 in the manuscript).

**Funding:** The author(s) received no specific funding for this work.

**Competing interests:** The authors have declared that no competing interests exist.

a comprehensive healthcare system and individuals' access to care is based on their ability to pay. Despite this wide range of models of healthcare, the World Health Organization (WHO) [4] has prioritised integrated care management (ICM) as its preferred policy approach. As a concept, ICM emerged over 20 years ago, and the history of healthcare research shows that there have been many attempts to achieve greater integration in provision since then [5]. As a concept, ICM prioritises the better integration of healthcare services to achieve enhanced patient care, seeking to reduce fragmentation. In theoretical terms, this approach adopts a more holistic view of the patient journey through a healthcare system than previous models, and should enable a more seamless experience of diagnosis, treatment, care, rehabilitation, and health for patients. The WHO prioritises ICM as a focus to support 'countries in moving their health systems towards universal health coverage, through equitable access to quality health services that are integrated, safe and people-centred across the care continuum' [6]. This people-centric approach shifts thinking away from an emphasis on the providers of complex healthcare systems [7, 8] towards enabling users to navigate their way through the system and receive seamless care by means of key touchpoints within that system that provide them with access to the treatment they need. Above all, ICM is predicated on a localised model of delivery designed to meet population needs, which, if effectively coordinated and implemented, can bring together diverse sectors, ranging from health and social care to mental health, primary care, and community services. As one of the largest publicly-funded healthcare systems globally, the National Health Service (NHS) in the UK (see Appendix 1), researchers have examined ways of seeking to innovate to build improvements to the delivery of services which have faced numerous financial and operational challenges.

In theoretical terms, achieving successful implementation of ICM as a concept poses a number of challenges in terms of its underlying integration objectives, which entail combining resources from multiple partner organisations, coordinating various services, and sharing information within a cohesive framework [14]. Here the challenge is in demonstrating the benefits of pooled resource use to achieve patient enhancements in a resource-constrained setting. The principles of ICM seek to facilitate a seamless transition of patients between different units, cultivating a comprehensive service that is tailored to specific patient groups' needs. This inclusive approach especially benefits individuals with chronic illnesses and mental health conditions, bridging gaps in care by consolidating services around the patient and facilitating effective information-sharing [15]. Ultimately, the aim is to elevate patient outcomes whilst enhancing the efficiency, quality, and overall effectiveness of healthcare services; the argument for the need to create more seamless delivery is strengthened by Allcock et al.'s [16] recognition that the major issue in NHS treatment is frontline delivery. Critically, numerous studies highlight the issues which are epitomised by Shaw et al. [17] that the initial challenge is one of recognising and harmonising the different paradigms that hospitals and healthcare managers adopt. These can range from a positivist, science-oriented approach that views the influences on healthcare as limited and manageable to social science approaches informed by a focus on human action and professional practice and that see far more complexity arising from the human experience of healthcare. Shaw et al. [17] point out that these healthcare environments are dynamic, ever-changing, and *messy* environments in which ICM is negotiated by different stakeholders and then evolves. Yet existing studies of efficiency and service delivery have tended to look at specific facets or elements of the hospital system as the literature review will show. The research gap that ICM implementation creates is in accommodating more holistic research approaches that are capable of reflecting this messiness and complexity whilst also incorporating diverse epistemological problems if we are to create ICM models that work in practice. What the gaps in knowledge demonstrate is that ICM models have to implicitly and explicitly recognise the different perspectives and influences

shaping stakeholder opinions around three key elements of ICM: structural, relational, and process-related change. Shaw et al. [17] used milestones and key events as a basis to measure the success of implementation, which NHS England [13] had also done, albeit using different nomenclature, to address the same questions around how innovations are adopted and diffused throughout an organisation. For this reason, the use of research tools that can accommodate the diverse views and positions of different stakeholders to improve delivery [13] is key in progressing system change. Yet the research tools also need to be capable of conveying the complexity of the hospital system in a meaningful way to illustrate how ICM and a seamless patient journey can be better achieved. One type of approach - albeit from a positivist tradition - that is capable of integrating a social science perspective at the research design stage to facilitate stakeholder input and engagement to effect system change in these messy and complex environments is the systems approach. Therefore, this offers a theoretically informed framework which has a practical and policy application to embed ICM as a new approach to patient care at a hospital level, with wider application in different contexts.

## Systems thinking and its application to health research

The systems approach, also known as a holistic view or systems thinking, recognises the interconnectedness of various sectors and actors and helps stakeholders understand how influences in one part of the system can impact the entire patient journey. The systems approach is derived from Operations Research (OR), which seeks to understand the totality of the situation (see [18] for a review of the evolution of the field and its focus on optimisation of resource use). Systems thinking as a dimension of OR approaches the issue of optimisation by examining the system in question (e.g., a hospital) through a lens of complexity as well as taking into account uncertainty, prediction, and interdependencies within the system so that critical relationships are understood. Among the underpinning concepts used is interconnectedness: it emphasises the importance not only of drawing together observations to create a synthesis but also of looking at the way in which interconnections are linked through feedback loops to try to identify relationships where causality exists. Different research methods from systems thinking have been deployed to approach healthcare systems, including dynamic modelling, agent-based modelling, causal loop diagrams, and other methods from social science such as social network analysis [19]. Most applied research applications of systems thinking in the health field typically commence with a discussion of the systems thinking theory; then the modelling is applied to a case study to demonstrate innovation or change that would lead to system improvement (see [20]). Among some of the obvious benefits of systems thinking in healthcare is its application to the analysis of workforce dynamics and of the value and effect of fostering collaborative coordination among partners to create joint solutions that mitigate impacts upon patient outcomes and care quality. Moreover, OR supports the optimisation of resource allocation, aiding policymakers in crafting inclusive policies that address shortages or service interruptions throughout the system by helping to recognise where breakpoints or bottlenecks exist. Through engaging stakeholders who are informed by differing paradigms and agendas, a systems approach helps identify how a comprehensive, collaborative, versatile, and flexible approach that cuts across boundaries and insular thinking may help address the myriad of challenges encountered within healthcare systems. However, the systems approach can be complex, and the available software is difficult for non-experts to use. Developing a decision support tool (DST) based on a systems approach not only makes the methodology more appealing but also enhances its practical usability. This enables end-users to interact with the system more effectively, thereby expediting the decision-making process. A DST in healthcare is a software program that aids key opinion leaders (KOLs) in making well-informed decisions by delivering pertinent and timely information. It consolidates data

from multiple sources, including patient records and medical research, to provide recommendations, predict outcomes, and propose best practices. These tools help in planning by identifying trends, optimising the use of resources, and enhancing patient care, which leads to more efficient and effective healthcare services.

This article, therefore, contributes to the development of research policies for organisations aiming to advance ICM by utilising a DST that facilitates better decision-making. It does so by building an innovative conceptual framework that uses systems approach principles to help with ICM implementation, using a simulation environment to grow and support integration-care-friendly policies. The purpose of this initiative is to demonstrate how the theory and practice of ICM can be invoked to bring about change and continuous improvements in patient care and outcomes through the use of innovative methodology in the form of a DST. Some of the key questions that need to be asked at the outset about the efficacy of using ICM in a hospital setting are framed in Table 1, which aims to help scope out what systems thinking may hope to achieve with its multidimensional and multidisciplinary approach to helping to improve the management of patient outcomes.

Table 1 presents a set of questions that serve as a guideline. These questions help the framework address a wide range of inquiries related to the implementation of ICM. Their global relevance lies in their ability to focus attention on the resources utilised—both financial and human—in the pursuit of patient-centred care and improved patient outcomes. To understand the broader system of healthcare delivery and the way ICM can be implemented, it is necessary to use systems theory to create a conceptual framework; this framework will then help to inform decision-making by depicting and quantifying the effects of any proposed changes in healthcare systems. The framework is a way to visualise and depict a complex system in a simplified manner, as many models in social science seek to do, but it differs from them in its adoption of a positivist approach. What makes a systems thinking approach fundamentally different is that this is not an abstract theoretical activity, as the framework is constructed only after detailed stakeholder engagement activities and dialogue have taken place to refine the system and adapt it to the local setting. Systems thinking is a powerful tool which enables stakeholders to have ownership over the visualisation process; it does so by helping to build in the local nuances and human dimension that positivist research methods have often been criticised as lacking.

In this paper, the conceptual framework was supported with a simulation environment that acted as a decision support tool (DST) to enhance retinal services at a prominent UK hospital. A DST is a software-based system that contains patient information from a multitude of

**Table 1. Questions on how operational research could achieve enhanced ICM in a hospital setting.**

| | |
|---|---|
| 1 | How can we ascertain the optimal and necessary composition and size of the workforce across integrated care settings in terms of optimising care delivery? |
| 2 | How does an integrated care approach impact waiting times across different healthcare services, and what policies and interventions can effectively minimise these waiting times within the integrated care system? |
| 3 | How does an integrated care model impact the management of chronic diseases across multiple healthcare settings, and what policies and interventions can be implemented or modified within integrated care systems to prevent or better manage chronic illnesses? |
| 4 | What policies and interventions can optimise efficiency, effectiveness, and cost-effectiveness whilst maintaining quality care within an integrated system? |
| 5 | How does integrated care influence health disparities among different demographic groups, and what policies and interventions within an integrated care framework can effectively mitigate these health inequalities? |
| 6 | How do the changes implemented within integrated care impact various health outcomes compared to traditional healthcare models, and what specific policies can be formulated within an integrated care approach to enhance overall health outcomes? |
| 7 | How does the implementation of community-based virtual clinics, independently and in combination with nurse-led services, affect patient waiting times and resource utilisation in hospital-based specialty services compared to baseline operations? |

sources, including details of individual conditions, previous interventions, and other scientific data that may help with clinical diagnosis and treatment. Alongside the DST is the care pathway, which is the specific continuum the patient follows after [21]. Vissers and Beech [22] identified five specific forms of healthcare planning whereby the care plan was a patient-focused element within the wider planning functions of healthcare management. These were:

- a care plan for each individual patient (patient planning and protocol);

- the planning of care in care pathways (patient group planning and control);

- the capacity planning of professionals, equipment, and space (resource planning and control);

- the planning of the number of patients to be treated and care activities to be carried out (patient volume planning and control), and

- the long-term policy of the institution (strategic planning).

Applying a simulation methodology makes it possible to combine many of these planning functions into a single system. This makes it possible to identify efficiencies and cost savings that could be made within the care pathway by integrating key services like community and primary care. It is also possible to target innovations at each level of planning, although we focus on the strategic level in this paper. Due to its holistic nature, systems thinking was identified in this study as a means by which the most effective and cost-efficient interventions could be identified in the hospital system to address the increasing demand for retinal services. The DST presented in this study constitutes a significant contribution to knowledge and was an innovative health research policy approach that was co-created with the hospital staff. It is significant because it provided key decision makers with evidence to help them prioritise those interventions that could lead to improved patient outcomes. The co-creation process is critical; NHS England [13] noted that system drivers (e.g., quality improvements) were insufficient to create change if they were not accompanied with staff engagement in co-leading the implementation of such innovations. The key communication process in systems thinking concerns illustrating the potential impact of interventions before practical implementation takes place, thereby informing strategic decision-making.

The paper commences with a discussion of the concept of ICM and its recent use to illustrate the theory and practice which it is embedded within. This is followed by a section on the application of systems thinking's contributions to achieving ICM objectives and its value in policymaking. We then introduce the conceptual framework we adopt for enhancing healthcare policies within the realm of integrated care. Next, we present a real case study, in which we developed a discrete event simulation model in collaboration with a retinal services unit at an NHS trust in England. The paper goes on to discuss the simulation's results and to draw conclusions from this example as regards the implications for ICM.

## Literature review

### ICM and its application in health-care management

Integrated care is a relatively new conceptual framework for healthcare management that has become a feature in several healthcare systems worldwide (see [17]), operating with varying degrees of partnership [23–25], but a formal inception at the national level is a relatively new ideology for the UK (see Appendix 1). The integrated care system (ICS) was formally established in July 2022 in England/UK [26]. This system consists of 42 geographically based partnerships, fostering collaboration among NHS service providers, commissioners, local

authorities, and other regional partners. It replaced the existing 106 clinical commissioning groups, a structure which had been criticised for its fragmentary nature. As a new framework for healthcare management, the potential effectiveness of ICSs at the national level remains to be seen [27], but many local success stories highlight their benefits, emphasising how they can be adapted to meet diverse needs. For example, in Derbyshire, the implementation of an integrated neighbourhood team approach resulted in a reduction of urgent ambulance callouts by 2,300 per year and further led to a decrease in hospital stays by 1,400 [28] (also see the NHS website for examples of successful changes https://www.england.nhs.uk/integrated-care/resources/case-studies/). As ICM is in its early stages in England, these ICSs are providing an important laboratory for research into how best to transition towards full integration. This process will require active participation from all levels of stakeholders and is described by NHS England [13] as pivoting upon motivating colleagues to embrace innovations, and implementing change management measures by means of shared and change leadership. As Thune and Mira [11] indicated, transitioning towards ICM requires involving all stakeholders in the process of designing, managing, implementing, and evaluating initiatives. Deriving empirical evidence of successful implementation and quantitative evidence of national outcomes whilst encouraged by WHO and OECD, it remains limited. As the evidence base to benchmark enhancements in patient-centred care are notably absent, van Harten [59] identified one example of breast cancer care and the qualitative changes in patient-centred care as well as the challenge of making comparative assessments. Zonneveld et al.'s [60] systematic review of the field reiterated these points, highlighting the values that were associated with behaviour change to achieve ICM. Clearly, this endeavour poses its own set of challenges. As already highlighted (e.g., [17]), organisational innovation is critical to integrate the wide range of stakeholders around a single model of delivery where diverse services and partnerships can be better coordinated by professionals and local partners who have a single goal - integration-friendly care that creates better health outcomes for the local population. Against that background, systems thinking has considerable potential to communicate the innovation proposed to a wide range of stakeholders and publics. It can do so by visualising and explaining the weaknesses in the current system in order to shift practice from a structure based on a complex amalgam of units and divisions towards a more harmonised model of delivery.

It is important to distinguish between the macro health policy objectives of government to improve citizen healthcare and the pragmatic requirements of local delivery, which are often focused on anchor institutions like hospitals. Whilst there is perennial criticism of the frequent policy changes at national level that have been made to try and address demand and supply issues for public healthcare across England (and the devolved nations), the most persistent criticisms of the NHS often stem from the fact that it consumes 20% of public spending to meet those needs [29]. Frequent policy changes have left local care in a constant state of flux over the last 20 years, as different governments have sought to reinvent the NHS and its interface with its publics. The current ideology of ICM has placed a considerable onus on hospitals to reorganise their delivery of services so as to address local concerns. These concerns typically revolve around historical criticisms and often unfounded assumptions about the bureaucracy and lack of joined-up thinking in the NHS [29]. Consequently, the national policy of pursuing ICM is the latest in a series of reforms that have impacted the NHS over its 77-year history, leaving frontline staff and managers to cope with constant changes over that period. Ideologically, ICM is premised on the notion that hospital services, as one facet of NHS delivery, are disconnected complex systems, and that this creates inefficiencies associated with poor resource use. As Wickens's [29] study illustrates, criticisms of NHS delivery are not necessarily borne out in practice, and delivery issues are not unique to the UK [30]. However, the new realities of ICS now mean that health managers need to embrace change once again and

adopt a new perspective on healthcare provision. That means those managers need to extend their purview beyond the confines of individual hospitals. ICSs must extend their modelling beyond hospital boundaries to encompass workforce needs, making sure the community dynamics, demographics, service demands, health and social care providers, mental health, and other services are combined in it.

## Systems thinking and the application to ICM

With the shift towards ICM and growing interest in how health systems operate, systems thinking has been widely used in healthcare as a method of real-world problem-solving (see Peters [31] for a review of the development of systems thinking, the theories underpinning its development, and the methods of analysis it deploys). As a tool designed to understand complexity in large systems like a hospital, it has great potential to help people understand how different components in a system can impact one another and how they are all related. This holistic approach contrasts with traditional linear problem-solving methods, offering deeper insights into system dynamics and interdependencies.

As a scientific research method, it has been adopted outside of academia to assist with policymaking. As Slater [32] explains in relation to the UK Civil Service:

*"Most policy problems occur within a complex system that is constantly changing with levels of uncertainty and a variety of cause-and-effect loops. Systems have multiple elements, interconnections, stakeholders and drivers, and stakeholders' views often do not align. Sometimes even defining the policy problem can be complex and hard to understand, let alone exploring possible solutions to that problem. Rather than understanding a system as the sum of its individual parts, systems thinking seeks to view every part as an element of the whole and it's the connections between these elements that are critical."*

Yet studies, such as Kwamie et al. [33] (p. 1715), have commented that policymakers have not fully recognised the importance of systems thinking. This oversight may stem from the inherent complexity and the required shift in mindset from viewing problems in isolation to considering them as part of a larger system. Peters [31], however, highlighted its wider applications in healthcare research. It would be suited to this field because of the very large complex organisations involved and the nature of the systems that have been created to manage public health from the primary care stage, e.g., Tako et al. [34] through to hospital systems. Simulation is a widely utilised approach within systems thinking, as it enables the system to be modelled and then experimented with to create different outcomes by using mathematical algorithms. Among the most notable methods used in healthcare research are discrete event simulation (DES), agent-based simulation (ABS), and system dynamics (SD), as well as hybrid variations of these techniques. The selection of methodology is heavily influenced by the objectives of the study [35]. For instance, when the study aims to analyse operational metrics at discrete time intervals (such as daily resource utilisation), DES is typically preferred [36]. Conversely, when evaluating actions (such as changes in the behaviour of medical staff and patients), ABS is more suitable [37], whereas SD is suited to strategic-level analysis regarding the causality and effects of relationships [38]. A comparison of the simulation methods is provided in Table 2.

DES is one of the most widely used system tools in a healthcare setting, as Zhang's [39] review documented. What is apparent from the wide application of DES in healthcare settings is the tendency for its use to focus on specific forms of treatment (e.g., Jahn et al. [40]), whether these be individual or multiple units (e.g., Huynh et al. [41]; Pendharkar et al. [42]), and its use as a means of examining how increased capacity and investment might improve waiting times. Other studies have included an analysis of a system of clinics offered in a particular hospital (e.g., Jun [43]) and examples of how patient flow can be improved through such

**Table 2. Comparison of simulation methods.**

| Simulation Type | Level | Components | Example of Application |
|---|---|---|---|
| Agent-based simulation | Behavioural | Action, interaction, agents (e.g., human) | To model infectious disease outbreaks [53] |
| Discrete event simulation | Operational | Events, Time, Entity (e.g., patient) | To capture patient flows in a clinic [54] |
| System dynamics | Strategic | Stocks, flows, cause and effect relationship | To evaluate the impact of policies on disease management [55] |

systems (e.g., Bard et al. [44]; Santos et al. [45]). However, a notable feature of the majority of the early studies of DES in healthcare settings is the reluctance to model whole hospital systems, an idea in which the concept of ICM is deeply embedded. However, a notable feature of early DES studies in healthcare is the reluctance to model entire hospital systems, where the concept of ICM is deeply embedded.

While these studies provide valuable insights, there is a need for a more critical examination of the assumptions and limitations inherent in these models. For instance, many DES applications focus narrowly on operational efficiency without fully addressing patient outcomes or systemic issues such as healthcare equity and accessibility. Additionally, the literature often highlights the technical aspects of simulation without sufficient consideration of the integration of care in which these healthcare systems operate.

## Materials and methods

Understanding the structure, inter-relationship, and dynamic behaviour of systems, and developing simulations for testing policies are some of the key features of system thinking [56]. The use of system thinking in practice improves healthcare settings, disease management, and public health [57–58]. This study chose to apply systems thinking because the recognised complexity of the hospital system made it necessary to achieve a comprehensive analysis of all of its elements and interdependencies as the system components.

System thinking approach and modelling involve several stages. In operationalising systems thinking, a series of steps are commonly followed: (i) system modelling, (ii) data collection and analysis, (iii) model development, and (iv) validation and verification [46]. These four stages are explained in the following sub-sections to provide guidance for applying the method.

### System modelling

The process of system analysis and modelling commences with the diagnosis of issues and the establishment of objectives to be pursued. Moreover, prior to conducting simulations, interventions or scenarios intended for testing must be clearly defined to allow seamless integration into the model. Additionally, the critical performance indicators expected from the simulation outcomes are identified during this phase. Depending on the project's complexity and overarching goals, the selection of an appropriate modelling or simulation methodology becomes paramount: possible options include DES, SD, ABS, and Monte Carlo simulation.

Qualitative mapping, also called conceptualisation, is the most important task in systems thinking and modelling. The task includes capturing and verifying patient flow across the providers (at the specialty level) and establishing key differences and features in a health system. Therefore, several meetings were organised with key stakeholders; this group consisted of key decision makers, the management team (hospital/specialty level), and key healthcare professionals. A focus group methodology was used to capture the free-flowing discussion around the visualisation of the system and its components and interdependencies.

To facilitate the assessment of healthcare policies pertaining to integrated care, we formulated a comprehensive conceptual framework. This framework encompasses primary care, secondary care, specialist services, mental health provisions, community services, involvement of local authorities, and engagement with other potential stakeholders. It is imperative to emphasise that our conceptual framework remains adaptable to the specific objectives and challenges encountered in each study. For instance, one study might require adding mental health services alongside hospitals, whilst another might prioritise community services working in conjunction with hospitals. Therefore, the framework can be customised to suit the specific needs and goals of each study, accommodating various simulation or modelling methodologies, and ensuring its applicability to diverse integrated care models.

Firstly, we present an example structure of an ICS in Fig 1. An ICS covers various catchment areas (called NHS trusts in the UK). Each catchment area hosts a variety of health and social care organisations, such as primary care, hospitals (including specialties), mental health services, and community services, as originally envisaged in the formation of the NHS. Secondly, Fig 2 illustrates the potential flow of patients through healthcare services within a catchment area, along with the structure of the key services available. The patient flow comprises healthcare services, both inpatient and outpatient, as well as primary care, community services, mental health facilities, and others.

Patients who are referred to inpatient and outpatient care wait for appointments within the community. Consultation, diagnostics, and treatment are processes in outpatient services; patients use them and are discharged the same day (i.e., no overnight stay is required) with the possibility of a follow-up appointment (if necessary). In inpatient services, which include diagnostics and surgeries, patients are often required to remain hospitalised (i.e., overnight stay); this can last up to several days, except for day-case admissions. Unplanned admissions to inpatient care may occur through direct referrals from emergency departments (EDs) or

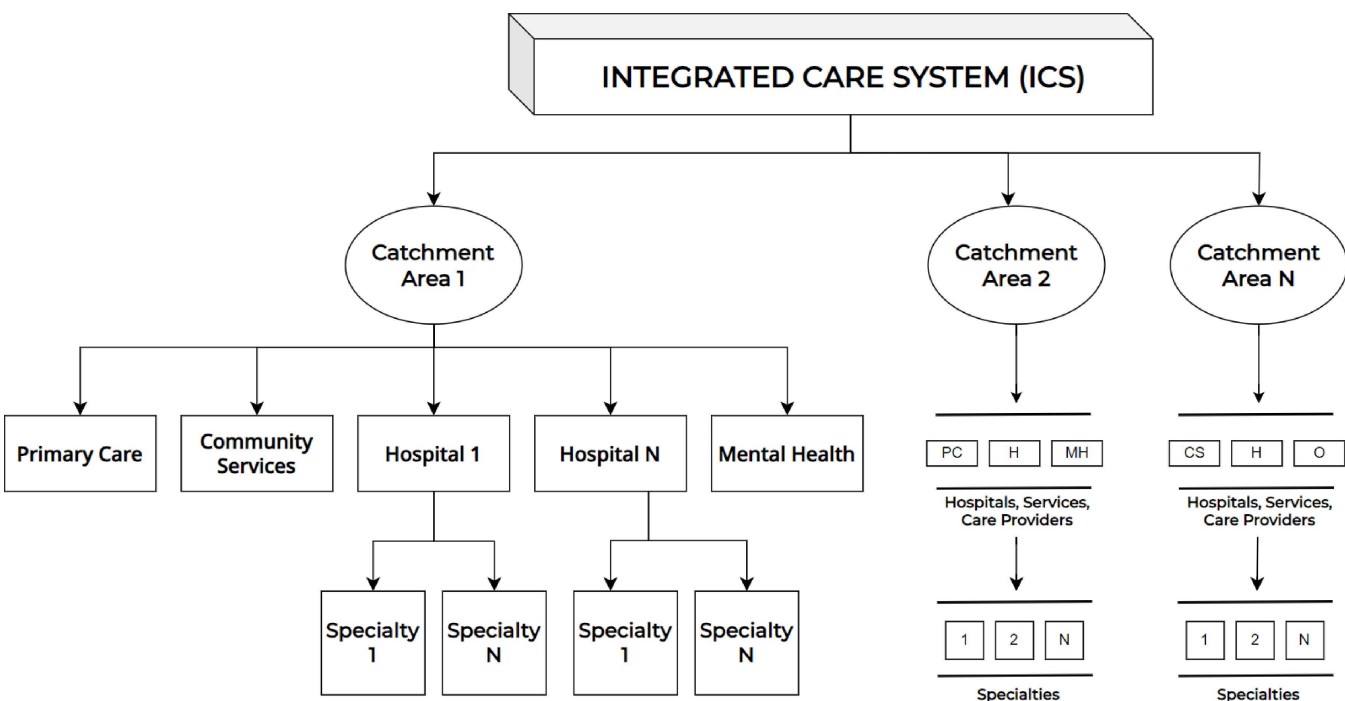

**Fig 1. An example structure of an integrated care system.** PC: Primary Care, H: Hospital, MH: Mental Health, CS: Community Services, O: Other.

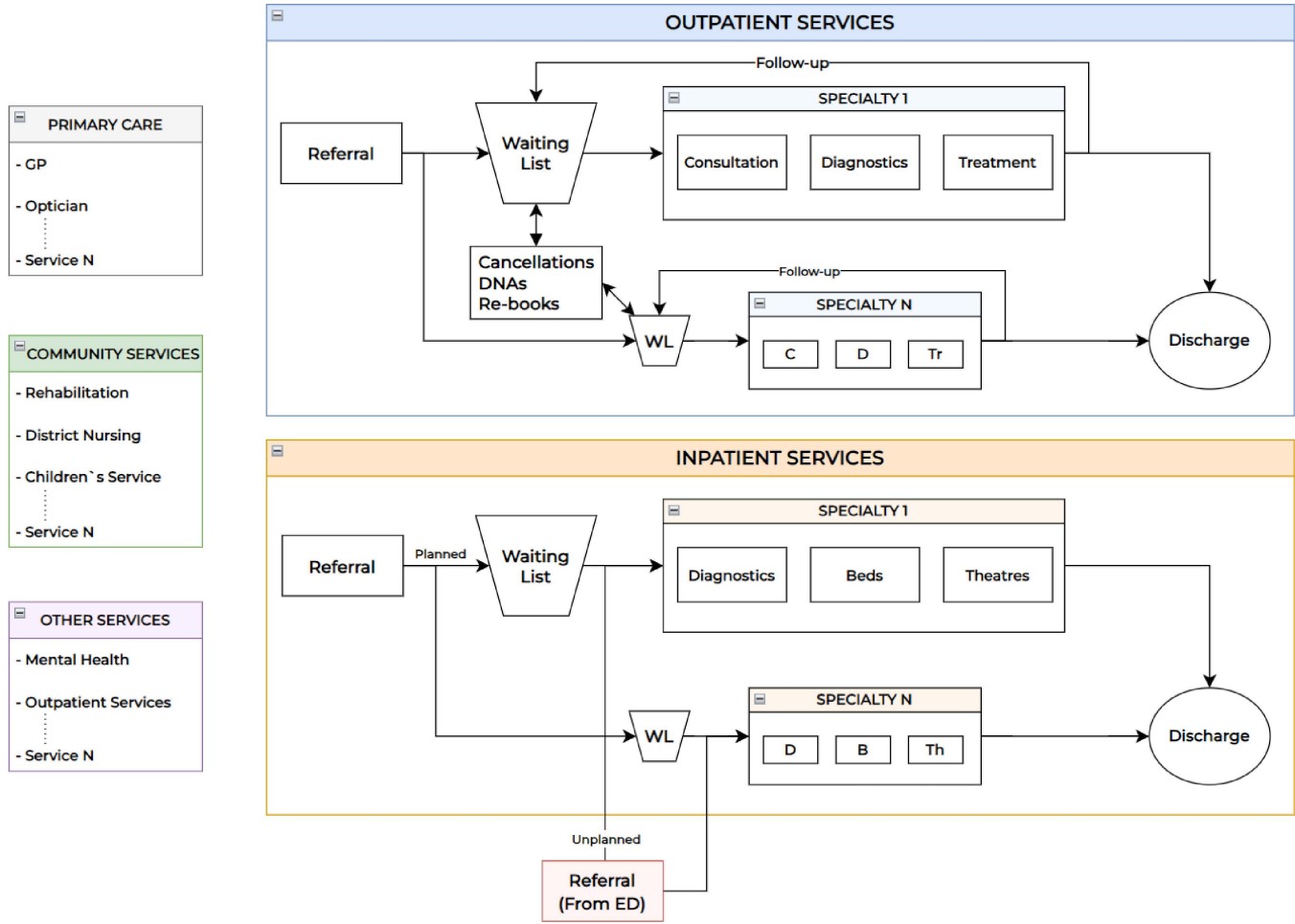

**Fig 2. Patient flow in healthcare services in a catchment area and the service structure.** GP: General Practitioner, DNAs: Did Not Attends, WL: Waiting List, C: Consultation, D: Diagnostics, Tr: Treatment, B: Beds, Th: Theatre, ED: Emergency Department.

other healthcare providers. Depending on the specific research objective and case study, the scope of EDs can be expanded as needed.

Patients receive examination and treatment in primary care settings, such as GPs' or optometry clinics. Community services comprise a wide area, including adult and children's services, various therapies (e.g., speech and language), and rehabilitation services (e.g., pulmonary or cardiac rehabilitation). Mental health services and outpatient care (e.g., physiotherapy) are also integral components. The seamless integration of these services is imperative, as they play a central role in delivering care, support, and preventive measures across health, social, and mental health domains. Without their inclusion, any attempt at an integrated care model would be incomplete. It is important to note that the patient flow in the real world is highly complex and transitions may take place between any combination of these services/departments. Thus, the conceptualised flow diagram is generic and adaptable to any setting, country, or integrated care service.

## Data collection and analysis

Conducting meticulous data collection and analysis is imperative for projects of this magnitude, particularly given the complexity of ICS. Such endeavours demand a substantial

investment of time and expertise due to the multifaceted nature of services which operate at an ICS level. Data for analysis may be sourced from national and/or hospital databases, published literature, and expert insights. The granularity of data can vary, encompassing patient-level or activity-level information, contingent upon the modelling methodology and scope of the study. Key input parameters typically include demand, capacity, referrals, wait times, treatment durations, length of stay (LoS), patient demographics, rates, cost-revenue considerations, and routing mechanisms. Statistical analysis is undertaken to examine the distribution of data (e.g., length of stay is a critical driver in the utilisation of resources) to inform modelling and simulation processes, with demand forecasting being necessary in certain instances.

## Model development

The subsequent step involves translating the flow or relationship diagram into a computerised environment, integrating established inputs derived from comprehensive data collection and analysis. The simulation may be developed utilising off-the-shelf commercial software (e.g., Simul8, Vensim, AnyLogic) or custom-coded through advanced programming languages, such as JAVA, Python, or R. As previously highlighted, the framework remains neutral towards specific simulation methodologies, refraining from endorsing any single approach. It is imperative to reiterate that the selection of the modelling method should be made at the outset, depending on the problem and the objectives of the study during the system modelling phase, as the subsequent mapping and data collection processes are intricately shaped by this initial decision.

This article presents a decision support tool (DST) specifically designed for retinal services. The tool's main objective is to evaluate the impact of interventions and policies at the operational level, for example, in terms of their impact on resource utilisation within the ICM framework. To achieve this, DES emerges as the preferred methodology. DES enables modelling of hospital operations and healthcare services, which in turn provides insights into patient pathways within the system and generating operational metrics for scenario analysis [47]. Therefore, DES was selected as the modelling approach on the grounds that it was the most suitable method for the case study.

It is important to reiterate that the conceptual framework is also usable with any other simulation method; the choice of method should depend on the project objective and focus. For example, if we wanted to study how the entire retinal services system changes over time (i.e., at the strategic level), including factors like improved vision and patient satisfaction, we would choose system dynamics simulation. However, in this case, we are focusing on specific events like treatments and resource usage in retinal services at discrete time points (at the operational level), rather than the overall system dynamics and feedback loops.

## Validation and verification

The model or simulation that is developed must undergo rigorous scrutiny to ensure its accuracy and reliability, a process commonly referred to as validation and verification. This entails scrutinising the logic of the model, as well as input generation to ensure proper functionality. The generated outputs are then meticulously compared against real-world data. Additionally, each component of the model is examined to confirm it accurately reflects the flow diagram. Crucially, key stakeholders within the system, who initially took part in the mapping process, are then asked to participate in validating whether the model accurately portrays the system's dynamics, in order to help co-create a robust real-world situation. Subsequent statistical analysis is conducted to determine the appropriate warm-up period and number of replications

required for simulation runs [48]. Once a validated and verified model is obtained, it can be utilised to conduct testing experiments and evaluate various scenarios. The model's outputs are then analysed to assess the potential impact of changes, guiding decision-making based on the analysis.

## Case study: Application of the model to an NHS hospital

The incessant strain on NHS services in the UK is a recognised feature that has severely impacted patient care and access, especially since the COVID-19 pandemic, as services grapple with a backlog on top of the existing demand. Evidence from September 2023 statistics reveals a staggering waiting list of 7.77 million patients: once patients awaiting more than one type of procedure have been accounted for, this amounts to roughly 6.5 million individual patients, with nearly 3.29 million of them enduring waits exceeding 18 weeks [49]. The second most affected service is ophthalmology: one in every 11 patients on the NHS waiting list, and almost 628,502 individuals, are awaiting these services [50]. Within ophthalmology, retinal services face the highest demand. This department treats conditions like age-related macular degeneration and diabetic retinopathy (DR), predominantly among the elderly population, through treatments like laser therapy or intraocular injections. Based on historical data from the NHS trust in this study, there has been a significant rise in DR follow-up attendances. Forecasts indicate this trend will persist, which will require retinal services to operate beyond 100% capacity. This situation leaves no space for new patients, and consequently, the waiting list and treatment times will be extended. The resulting delays in treatment could potentially lead to vision loss.

**System modelling in the case study.** To showcase the use of the conceptual framework to improve the system, we focused on retinal services as a case study. Semi-structured interviews and focus group meetings were carried out with specialist nurses from retinal services and managers from an ICS. The interviewees' insights and inputs allowed us to map the patient flow in the service in detail as well as the structure of the ICS. Also, inputs such as resource, capacity, and patient routing were collected via the interviews. The patient flow was verified by the nurses from the services.

In collaboration with the trust and using the conceptual framework as a guide, we developed a DST for the entire patient pathway of retinal services. This tool integrates the primary care, hospital, and community services responsible for comprehensive care for individuals with eye-related conditions. The goal was to identify the most effective and efficient interventions to alleviate the pressures on retinal services.

Based on the conceptual framework, a diagram of retinal services, showing the structure of services and the hierarchy, is represented in Fig 3. It consists of two layers. A high-level depiction of retinal services and its hierarchy under an ICS is shown in part (a) of Fig 3. The patient flow for retinal services was conceptualised by the trust's stakeholders, including the service manager and nurses. Part (b) of the diagram shows diagnosis, treatment, and consultation, with the possible options listed under each stage.

Fig 3. and other parts of the DST are established through a co-creation process, which involved extensive collaboration with key stakeholders from the Trust's retinal services, including consultant ophthalmologists, specialist nurses, and the service manager. This engagement was structured through three comprehensive rounds of semi-structured interviews and focus groups, each serving a distinct purpose in the model development process. University ethical approval was obtained for this study. For all interviews and workshops, formal participant consent was obtained from healthcare professionals, with clear information provided about data usage, storage protocols, and confidentiality measures in accordance with institutional requirements.

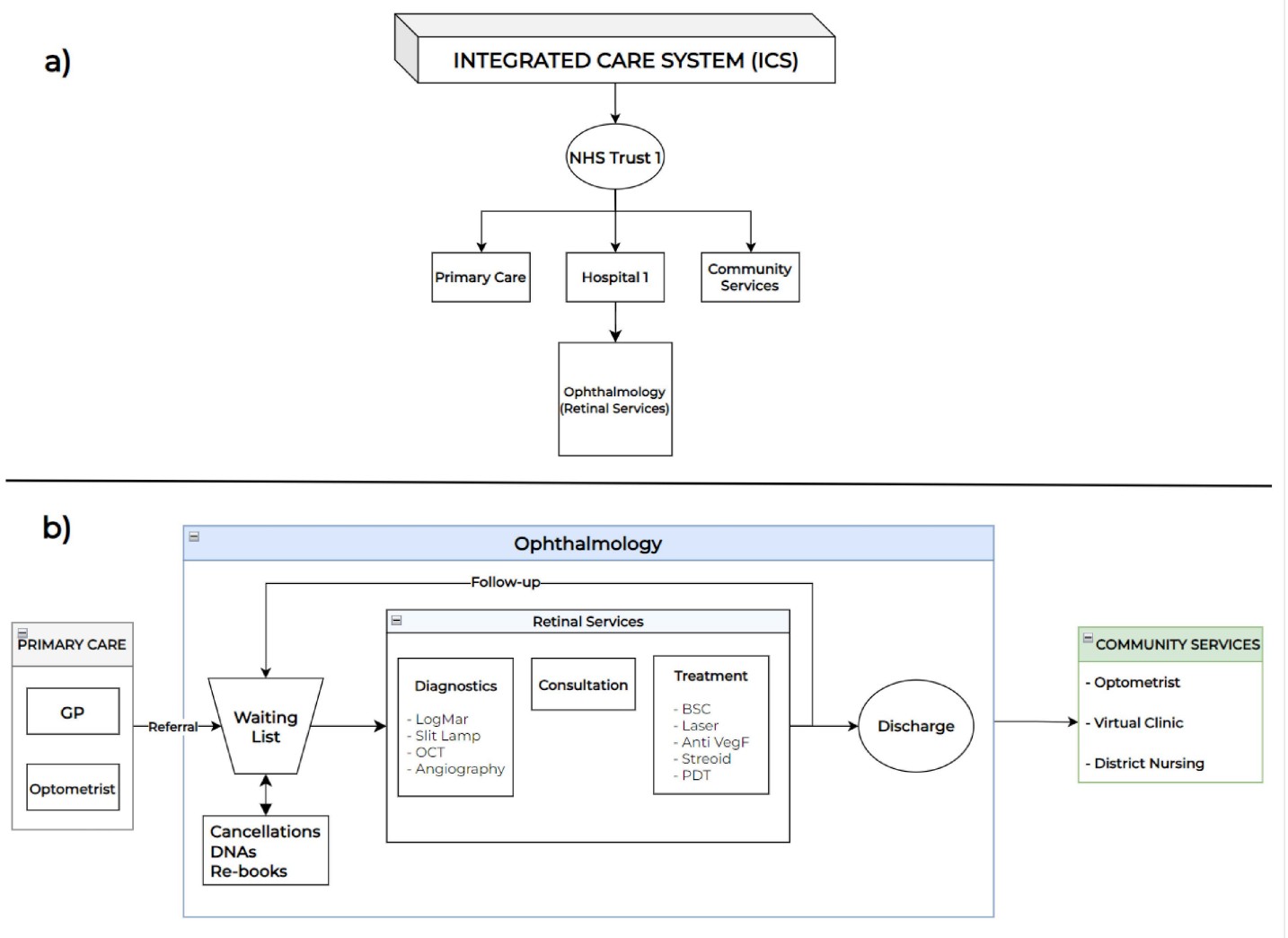

**Fig 3. Conceptual diagram of retinal services.** a) High-level depiction of retinal services and its hierarchy under an integrated care system; b) Patient flow in retinal services at trust level.

The first round focused on understanding and mapping the current patient pathways. Through semi-structured interviews, we explored:

- What is the complete patient journey from referral to discharge?

- What are the key decision points throughout the pathway?

- What resources (human and equipment) are required at each stage?

- What are the typical treatment times for different procedures?

- How do patient flows vary between different conditions and treatments?

The second round concentrated on verification and detailed data collection. Stakeholders were presented with initial pathway mappings and asked to verify and enhance them. Key questions included:

- Are these pathway representations accurate and complete?

- What specific resources are consumed at each touch-point?

- What variations exist in treatment times between different patient types?

- What factors influence clinical decision-making at each stage?

- What are the current bottlenecks and resource constraints?

The final round focused on model validation and scenario development. Stakeholders were presented with the complete pathway mapping and preliminary Simul8 model. Questions explored:

- Does this model accurately represent your service operations?

- Are there any missing elements or inaccuracies in the model?

- What scenarios would be most valuable to test?

- What metrics would be most useful for evaluating different scenarios?

Each round of stakeholder engagement led to specific refinements in the model. The iterative nature of this process ensured that the final simulation model accurately represented the real-world operation of the retinal service. All stakeholder interactions were thoroughly documented through meeting minutes, pathway diagrams, and data collection sheets, ensuring transparency and traceability in the model development process. This structured approach to stakeholder engagement was crucial in developing a model that not only accurately represented the current service but also enabled meaningful scenario testing for service improvement.

Patients are mostly referred to these services via optometrists and GPs. After the initial diagnosis and treatment, patients either attend for the remaining treatments (e.g., injections) or a follow-up review (monitoring/assessment). Treatment and follow-up patterns vary depending on patient type and the type of condition: namely, age-related macular degeneration (AMD), diabetic retinopathy (DR), or retinal vein occlusion (RVO) conditions. Re-bookings can be made for patients who cancelled or 'did not attend' (DNA) appointments.

**Data collection and analysis in the case study.**  To run the simulation model for the case study, we collected various data which was mainly patient level. The main data sources were the hospital, expert views and published literature/reports. The data obtained from the hospital were from a period of 36 months. The process involved comprehensive data collection and analysis due to the large size and complexity of the dataset. Working closely with the hospital's health informatics team, we extracted relevant data from various services, including inpatient, outpatient, and emergency services. This data was used to estimate various input parameters, establish statistical distributions, and forecast future demand. Also, several statistical distributions were set to capture the variation in real life for factors such as length of stay and waiting time.

The full list of the input parameters that are used in the model is provided in Table 3 along with the source. The inputs are related to demand, human and non-human resources (such as ophthalmology consultants, nurses, and diagnostic equipment), routing rates, and treatment rates, as well as costs and revenues.

**Model development, validation and verification in the case study.**  Forecasting algorithms were used to determine the 12-month demand for each patient type (AMD, DR, and RVO). To forecast trends for the next 12 months using statistical approaches, we partitioned the data into age groups, further broken down into first appointments and follow-up, and applied various algorithms such as ARIMA, exponential smoothing, and multiple linear regression to forecast each age group. Subsequently, we selected the algorithm that

**Table 3. Input parameters.**

|  | Estimate | Distribution | Source |
|---|---|---|---|
| **Demand** | | | |
| Number of monthly first appointments | Forecasted | N/A | Hospital data |
| Number of monthly follow-up appointments | Forecasted | N/A | Hospital data |
| Number of monthly did not attends | Forecasted | N/A | Hospital data |
| **Treatment** | | | |
| Number of follow-up appointments for AMD patients in each treatment year | First year: 5<br>Second year: 4<br>Third year: 4 | Log normal | Hospital data |
| Number of follow-up appointments for DR patients in each treatment year | First year: 4<br>Second year: 3<br>Third year: 3 | Log normal | Hospital data |
| Number of follow-up appointments for RVO patients in each treatment year | First year: 4<br>Second year: 3<br>Third year: 3 | Log normal | Hospital data |
| % of patients for each treatment year | First year: 72%<br>Second year: 18%<br>Third year: 10% | Multinomial | Hospital data |
| % of patients discharged at the end of year 1, year 2, and year 3 of treatment | First year: 5%<br>Second year: 2%<br>Third year: 1% | Multinomial | Hospital data |
| **Diagnostics** | | | |
| % of patients using the following diagnostics | LogMAR: 100%<br>Slit lamp: 100%<br>Angiography: 20%<br>OCT: 70% | Multinomial | Hospital data |
| Available number of diagnostics | Slit lamp: 2<br>Angiography: 1<br>OCT: 2 | Fixed | Hospital data |
| **Cost - Salary** (Hourly) | | | |
| Consultant | £48.64 | Fixed | PSSRU |
| Ophthalmic photographer | £17.66 | Fixed | PSSRU |
| Nurse | £14.27 | Fixed | PSSRU |
| Technician | £14.27 | Fixed | PSSRU |
| Healthcare assistant | £9.85 | Fixed | PSSRU |
| Optometrist | £17.66 | Fixed | PSSRU |
| **Revenue** | | | |
| 1st appointments | £112 | Fixed | National Tariff Payment |
| Follow-ups | £63 | Fixed | National Tariff Payment |
| Laser | £292 | Fixed | National Tariff Payment |
| Photo dynamic therapy | £107 | Fixed | National Tariff Payment |
| Best supportive care | £112 | Fixed | National Tariff Payment |
| **Resources** | | | |
| Number of available clinics/week | 33 | Fixed | Hospital data |
| Number of ophthalmic photographers | 1 | Fixed | Hospital data |
| Number of nurses | 4 | Fixed | Hospital data |
| Number of technicians | 2 | Fixed | Hospital data |
| Number of healthcare assistants | 5 | Fixed | Hospital data |
| Injection bed | 1 | Fixed | Hospital data |
| Theatre | 1 | Fixed | Hospital data |
| Consultation room | 4 | Fixed | Hospital data |

AMD: age-related macular degeneration; DR: diabetic retinopathy; RVO: retinal vein occlusion; PSSRU: Personal Social Services Research Unit

best aligned with the data patterns and incorporated it into the simulation as demand-related inputs. This process yielded an exceptionally accurate forecast for the hospital.

Based on the simulation, retinal services anticipate an average of 94 new patients (referred to as first appointments in the NHS) and 1088 follow-up attendances per month for the financial year spanning from 1 April 2023 to 31 March 2024. This estimation is based on forecast activity, with monthly expectations ranging between 79 and 113 new patients and 939 and 1178 follow-ups, with an average new-patient-to-follow-up ratio of 12.

Next, the model was developed using commercial simulation software (Simul8). The conceptual diagram was transferred to the computer environment which then fed by the input parameters. The model was meticulously validated and verified by following the procedures described in the previous section, with the involvement of stakeholders from the service. The stakeholders confirmed that the model accurately represented real-life service flow. Key outputs of interest in validating the simulation model included the number of activities (categorised by patient and appointment types), utilisation rates, costs, and revenues. The simulation demonstrated high accuracy, with differences between results and observed data within a 5% margin.

**Scenarios and interventions in the case study.** After discussions with experts such as consultants, nurses, and service managers, three interventions, along with a baseline scenario, were identified as effective ways to alleviate pressure on retinal services. These scenarios were as follows:

1) Baseline scenario (SC0): This involves the status quo for retinal services, without altering any existing operational procedures or patient flow.

2) Nurse injectors with a 5% increase in DR arrivals each month (SC1): This involves utilising and training existing non-medical practitioners, such as ophthalmic clinical nurse specialists, to administer intravitreal injections, and increasing monthly DR arrivals by 5%, thereby alleviating the workload of ophthalmology consultants and referral of new patients.

3) Community virtual clinic (SC2): This involves virtual follow-up assessments where patients are seen by a trained nurse instead of the medical team, thereby alleviating pressure on all aspects of retinal services, including consultants, clinics, and diagnostic services. This will apply to patients who are stable after treatment or without a treatable disease, but ongoing monitoring is required.

4) Nurse injectors with a 5% increase in DR patients and community virtual clinics (SC3): this involves the implementation of SC1 and SC2 simultaneously.

The next section evaluates the scenarios and analyses the model outputs.

## Results

The retinal service simulation tool is adaptable and can be tailored to different situations, allowing for the evaluation of various policies. Together with the baseline, the four scenarios described above were run in the simulation. These interventions underwent further testing to thoroughly evaluate their impact on activity and resource utilisation.

Figs 4 and 5 depict the average estimates spanning a 12-month period from April 2023 to March 2024. Whilst detailed monthly outputs exist for 44 key performance metrics (e.g., activity, resource utilisation, and diagnostic and treatment procedures) comparing two scenarios in each run, we have condensed the selected key performance indicators (KPIs) for easy comprehension among key opinion leaders, facilitating quicker decision-making.

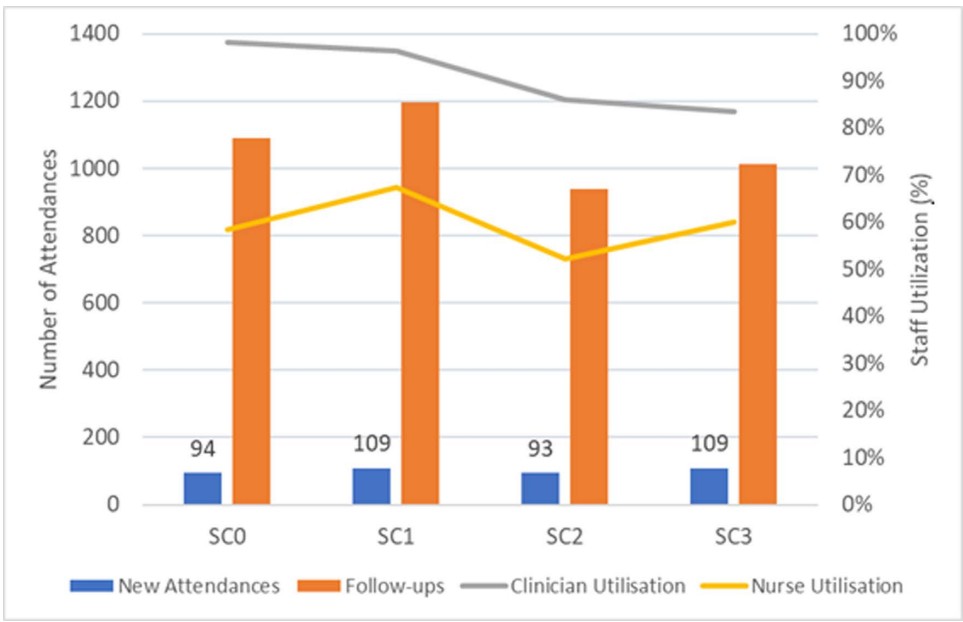

**Fig 4. Evaluating the activity and staff utilisation implications of SC0, SC1, SC2, and SC3.** SC0 = Baseline scenario; SC1 = Nurse injectors + 5% increase in DR arrivals each month, SC2 = Community virtual clinic, SC3 = Nurse injectors + 5% increase in DR patients and community virtual clinics.

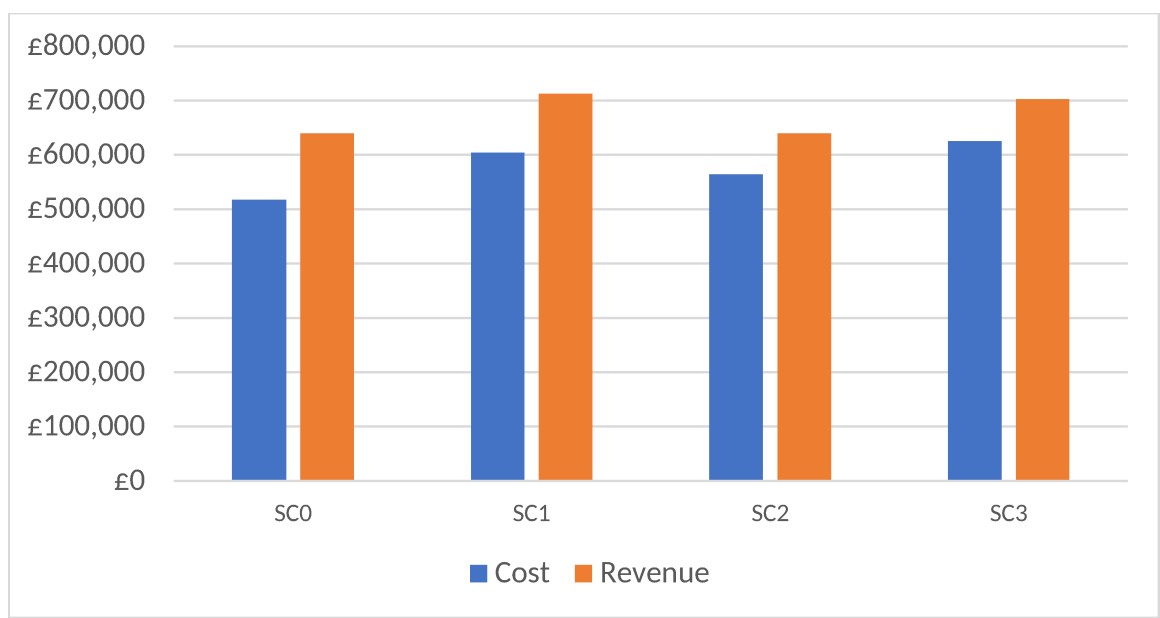

**Fig 5. Evaluating the financial implications of SC0, SC1, SC2, and SC3.** SC0 = Baseline scenario; SC1 = Nurse injectors + 5% increase in DR arrivals each month, SC2 = Community virtual clinic, SC3 = Nurse injectors + 5% increase in DR patients and community virtual clinics.

According to the simulation results, combining nurse injectors with a 5% increase in DR arrivals each month (SC1) and adding a community virtual clinic (SC3) result in the highest increase in new attendances. This shows a 16% increase compared to the baseline scenario, which involves no changes to retinal services. Meanwhile, SC1 demonstrates a 10% increase in follow-ups. Whilst SC1 alleviates the pressure on the waiting list for initial referrals, it elevates the number of follow-ups without adding an undue burden on clinicians. Specialist nurses handle a certain number of injections, resulting in a marginal 2% decrease in clinician utilisation, from 98 to 96%. Notably, nurse utilisation rises to 67% in SC1, up from a baseline of 59%.

In SC3, an increased use of virtual clinics in the community notably impacts nurse utilisation, also within reasonable limits, whilst ranking as the second-best option for clinician utilisation. The scenario increases monthly new attendances by implementing a 5% monthly increase in initial referrals and reducing follow-ups from 1,088 in Scenario 0–1,012 in SC3.

Consequently, an average of 76 patients per month, selected for routine check-ups based on clinical stability, would be attended by nurses operating in the virtual community clinics. Due to the implementation of SC1 and SC3, whereby more patients are referred for diagnosis and treatment, we expect a gradual 5% decrease in overall primary care appointments related to retinal services over time. This reduction will alleviate some pressure on primary care GPs, who are already under significant strain.

Fig 5 highlights that SC3 is associated with the highest monthly cost and revenue. This is attributed to an increase in new patients, which results in approximately 21% higher costs and 10% higher revenue compared to the baseline. These figures include the expenses related to establishing virtual community clinics, such as telehealth platforms and remote monitoring tools, as well as all other costs associated with diagnosis, treatment, and staff.

These results enable the evaluation of changes at a broader system level rather than in isolation. This approach aligns with the objectives of integrated care, which aims to bring together local partners and services—such as primary, secondary, and community care in the case of retinal services—to improve local patient services. By introducing virtual community clinics and nurse injectors, new referrals can increase, leading to early intervention that is known to improve vision and outcomes. Additionally, examining the impact on the broader system aids in reducing disparities in outcomes, patient experiences, and access to care. This comprehensive analysis also empowers key opinion leaders to improve productivity and cost-effectiveness, thereby aiding the NHS in fostering wider social and economic growth.

## Discussion

The adoption of ICS is now expanding, in response to both state policy and demand pressures; this necessitates the implementation of research policies designed to co-create solutions that will help map frontline delivery to ICM ambitions. However, the implementation of ICM has been made subject to further demand pressures as a result of the complex and numerous networks of organizations involved in frontline delivery. Whilst the ICM policy agenda set at a national scale has laudable philosophical objectives for patient care, an understanding of patient pathways suggests that a step change in healthcare modelling will be needed to elevate our thinking away from single-service delivery. Instead of focusing on single services in isolation, the modelling needs to embrace whole-system approaches in which the ICS are embedded; this, in turn, means adopting novel research policies that enable us to co-create a holistic perspective on service delivery. In the wider services management literature this approach has largely been associated with the service blueprinting methodology [51], which seeks to understand where service innovation and enhancement could occur. The strength of the DST is that it takes us further than simple service innovation based on service design/redesign, as

it creates a quantified model of how human actions may impact service delivery in messy, real-world environments [17]. Our conceptual framework offers numerous advantages for key opinion leaders (KOLs), including:

1. **Workforce optimisation,** which determines the ideal workforce composition for integrated care settings so as to enhance care delivery efficiency;

2. **Innovative service delivery**, attained by developing policies and interventions that optimise efficiency, effectiveness, and cost without compromising on quality of care;

3. **Community health outcomes,** which can be positively influenced by identifying integrated care strategies that address health disparities; community-based interventions targeting specific groups can mitigate inequalities and enhance outcomes;

4. **Resource planning,** which facilitates assessment of non-human resources like bed, clinic, and theatre capacity so as to meet existing and future demand more effectively;

5. **Capacity assessment,** whereby through identifying those treatment pathway elements with capacity gaps, service innovations to address those gaps can be proposed and evaluated;

6. **Minimising waiting times:** these waits can be effectively reduced by means of policies and interventions implemented by individual hospitals and their clinics;

7. **Chronic disease management**, which entails identifying policies and interventions that can help prevent or manage chronic illnesses within ICSs.

Our study integrates systems theory within the healthcare context, providing a comprehensive model for understanding and managing complex adaptive systems. By combining hospital and out-of-hospital services into a single framework, we advance the theoretical understanding of systems integration in healthcare. This approach underscores the importance of considering the interdependencies and interactions between different components of the healthcare system, which is crucial for developing effective ICMs. Additionally, the development and application of DSTs in our study contribute to the theoretical discourse on decision-making in healthcare. The DST empowers KOLs to interconnect service demand, resource utilisation, and operational requirements within integrated care settings so that they can then make critical decisions on priorities and resource use. The scenario-planning options within the DST are tailored to local populations, so as to more accurately forecast both their current and future operational and financial needs. This approach addresses distinctive challenges using a holistic perspective that varies across settings, countries, and healthcare systems; hospitals can apply this tool across all their services to help them make management decisions regarding how to deploy resources for best effect. This case study, based on a co-creation model of collaboration with an NHS retinal service at an English hospital, showcased how the conceptual framework can be used in practice, and identified three interventions to alleviate pressure on hospital clinicians (ophthalmology consultants) and GPs in primary care settings.

Simulation-based comparisons using the DST revealed significant outcomes. SC1 (nurse injectors + 5% monthly referral increase) indicated a 16% rise in new attendances and a 10% increase in follow-ups. This scenario effectively reduced waiting lists without overwhelming clinicians. On the other hand, SC3 (combining SC1 with community virtual clinics) notably affected nurse and clinician utilisation. Implementing SC3 resulted in a 21% cost increase and 10% higher revenue due to increased patient numbers and the establishment of a virtual clinic. Overall, SC1 and SC3 foresee a gradual 5% reduction in primary care appointments, potentially easing the burden on GPs. In response to these findings, the outpatient transformation lead commented: '*without the simulation model, it would have been impossible for us*

*to recognise that such a simple intervention could lead to a significant impact on our services. The ophthalmology simulation model promoted the ophthalmic team's understanding of forecasting and interventional effect, which has been very helpful for our department in planning service developments to meet demand*'. The DST has positively influenced the decision-making process, presenting opportunities to enhance patient outcomes, service efficiency, and staff satisfaction in all areas of hospital service provision.

However, the success of implementing virtual clinics is contingent upon having a strong digital infrastructure and ensuring that patients have access to the necessary technology, which may not be consistently available across all NHS regions. This discrepancy could worsen existing health disparities, particularly in rural or underserved areas. Furthermore, the introduction and training of nurse injectors in retinal services demand substantial investment in both education and continuous professional development to maintain high standards of patient care. There may also be resistance from current staff and potential regulatory obstacles that could hinder the quick adoption of these new roles. Additionally, the effectiveness of virtual clinics and nurse injectors relies on smooth coordination and communication across various levels of care, which is often challenging within the NHS's complex and fragmented system.

While previous studies have explored DES in healthcare settings, our work presents the first comprehensive DES model specifically designed for healthcare services within an integrated care framework. For instance, unlike existing models that typically focus on isolated aspects of ophthalmology services [61], our approach uniquely captures the entire patient pathway, incorporating multiple service points, resource interactions, and care transitions within a single modelling framework. This holistic approach, embedded within a DST environment, enables system-wide analysis and optimisation that was not previously possible with more narrowly focused models. This comprehensive integration of services within a single simulation framework represents a novel contribution to both healthcare modelling and ophthalmology service improvement literature.

It is important to consider that these interventions hold potential for application in diverse healthcare settings or countries with comparable healthcare systems. For instance, in countries like Canada, Australia, and parts of Europe, where integrated care is prioritised amid challenges of rising demand and resource constraints, similar benefits may be realised. However, the success of these interventions in varied contexts hinges on several critical factors. Challenges observed in the NHS, such as digital infrastructure limitations, investment needs for nurse injector training, and the necessity for effective coordination across care levels, are applicable globally. Moreover, differences in cultural norms and regulatory frameworks in healthcare delivery systems necessitate careful consideration. Adaptations tailored to local practices are essential to ensure these interventions effectively address specific healthcare system needs and conditions. Therefore, while the fundamental principles of these interventions are broadly applicable, their implementation requires customisation to optimise outcomes within each unique healthcare setting.

Systems approaches are gaining awareness and popularity in addressing complex, messy systems through the use of simulation-based models (e.g., SD, DES, and ABS) to establish the most efficient and effective delivery of services. However, there are limited examples specifically developed for the context of ICMs. Numerous reports, studies, and research stress the need to integrate fragmented services to improve the health of the local population. Harten [52] explicitly discusses the benefits of ICMs and highlights that a firm body of evidence on the added value of transforming pathways into ICMs is hard to find, leaving room for much variation. This article addresses that major gap in the literature by providing evidence for KOLs before such changes are implemented in practice.

Like any research study, our framework has limitations. Whilst integrated care settings differ across global healthcare systems, our conceptual framework might not suit every organisation that works with partners within an integrated care framework. The versatility of this framework allows for seamless adaptation to diverse integrated care settings worldwide, ensuring alignment as needed. Its comprehensive, step-by-step guide to constructing models will ensure it can be applied using a structured and systematic approach.

The complexity of stakeholder engagement across multiple organisations also presented challenges. While we engaged with key stakeholders from the retinal services, achieving comprehensive representation from all touchpoints in the patient pathway (including primary care and community services) proved challenging. This limitation potentially affects the model's ability to capture all nuances of inter-organisational interactions and their impact on patient flow. Furthermore, stakeholder availability constraints meant that some validation sessions had to be conducted with partial representation, which could introduce bias in the model's assumptions and structure. The varying levels of technical expertise among stakeholders also influenced their ability to critically evaluate certain aspects of the model, particularly its technical components.

Because of the intricate nature of integrated care, in which multiple organisations collaborate in treatment delivery, the patient pathway often becomes very complex, requiring many input parameters and statistical distribution determinations, such as LoS and waiting time distributions. This complexity presents two further limitations: first, extensive data requirements pose a challenge, especially in healthcare settings lacking in-house electronic health record systems. Collecting data at this scale might not be viable due to resource limitations and challenges when it comes to accessing the necessary data. For example, precise treatment times, resource utilisation rates, and detailed patient pathway variations are rarely captured systematically. This limitation becomes more pronounced when modelling integrated care pathways that span multiple organisations and care settings. In our study, while we obtained core operational data from the retinal services, granular data about patient transitions between services, detailed resource consumption patterns, and accurate waiting times at various pathway stages were not readily available. This challenge of data availability often necessitates extensive manual data collection, expert estimations, and assumptions, which could impact model accuracy and generalisability.

Second, developing the DST necessitates a high level of skill in, among other proficiencies, qualitative mapping, statistical modelling, health economic analysis, and simulation modelling expertise. Many health systems lack the requisite expertise and funding for the development of such tools. Future research will focus on overcoming the second of these two limitations by creating a universal web-based DST, hosted on the cloud. This user-friendly system will incorporate advanced data visualisations and offer easy customisation of various integrated care settings. It will feature customisable input parameters, making it possible to adapt the tool to local contexts, and it will be possible to carry out instant policy comparisons against baselines through simply clicking a button.

## Conclusions

This study has emerged from the urgent need for innovative research policies to be developed that will help the NHS address real-world problems of ICM and demand management in hospitals. In focusing on an ICS, the study aligns with the new policy environment that promotes ICM with the aim of innovating to effect change in the NHS, thereby enhancing patient outcomes [13]. What the research demonstrates is that even where national policy frameworks have established a new approach to service delivery, research tools are needed

(e.g. DSTs) at a local level to assist in achieving national ambitions to optimise healthcare services. The use of systems thinking moves national policy nearer to implementation at a local level, in keeping with NHS England's change agenda [13]. For individual hospitals, this study and its outcomes offer policymakers and KOLs an evidence base which can help them make more rational and inclusive decisions on the allocation of resources and expertise necessary for the development and deployment of universal, user-friendly DSTs. Not only does this study offer evidence of the value of innovation and the cost-benefit of any investment decisions; at a practical level, but it also assists in bridging resource gaps, encouraging standardised data collection methods, and investing in enhancing those skill sets that are essential for DST utilisation. As the landscape of integrated care evolves, policymakers have the opportunity to shape the future by supporting initiatives that streamline the adoption and utilisation of DSTs. This alignment between policy and technological innovation offers considerable potential for transformative changes in healthcare systems that could ultimately provide benefits in terms of both patient outcomes and service efficiency. As Allcock et al. [16] advocated, accelerating change in the NHS requires innovation in frontline delivery; this study demonstrates how the diverse actors within hospitals can coalesce around one such innovation, one that is systems focused but that also connects many of the agents together. That innovation seeks to embody the common aspiration of stakeholders to enhance patient care so that there is a greater degree of team and system cohesion. That can be achieved by identifying how simple changes, evaluated through key events and milestones, can make a significant difference in the implementation of ICM [17].

## Appendix 1: The national health service in the UK

In the UK, the initial model for the provision of services was set out in the National Health Service (NHS) Act (1946, A3) to 'promote the establishment in England and Wales of a comprehensive health service designed to secure improvement in the physical and mental health of the people of England and Wales and the prevention, diagnosis and treatment of illness, and for that purpose to provide or secure the effective provision of services'. At the heart of the NHS system when it was founded were three interconnecting elements: the primary care model (e.g., General Practitioners (GPs) as gatekeepers to other services like hospital care); community services (e.g., home nurses, public and environmental health) and a network of state-owned hospitals (historic models of mental health treatment and care of the elderly were eventually integrated into the system in the 1990s). Initially, each of these service elements was managed separately. Over time, the effectiveness of this model of delivery was periodically called into question, as the scope and extent of service provision continued to expand; periodic remodelling and reform took place to reconfigure services [9,10]. Thune and Mina [11] make a compelling case for a focus on the hospital in the healthcare system for the reason that it is a central actor in health innovation, particularly in the adoption of new technology and as a context for the creation and recipient of organisational innovation. As hospitals have evolved since the 1940s into a complex interconnected system for healthcare delivery, with a multitude of different functions and processes, each with interdependencies, they are an important laboratory in which innovations to improve performance can be developed and implemented. As Scheinker and Brandeau [12] caution, however, the success of innovations ultimately hinges upon stakeholder engagement, technical performance, implementation, and sustained use. The national management of the NHS [13] works to promote constant change and local innovations that enhance patient care, alongside engaging in national policy initiatives and periodic restructuring, such as the implementation of ICM [10].

## Acknowledgements

None

## Author contributions

**Conceptualization:** Eren Demir, Usame Yakutcan, Stephen Page.

**Data curation:** Eren Demir.

**Formal analysis:** Eren Demir, Stephen Page.

**Investigation:** Eren Demir.

**Methodology:** Eren Demir, Usame Yakutcan.

**Project administration:** Eren Demir.

**Software:** Eren Demir.

**Supervision:** Eren Demir, Stephen Page.

**Validation:** Eren Demir, Usame Yakutcan.

**Visualization:** Eren Demir.

**Writing – original draft:** Eren Demir, Usame Yakutcan, Stephen Page.

**Writing – review & editing:** Eren Demir, Usame Yakutcan, Stephen Page.

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
