## [Decision Letter · Decision Letter 0]

28 Jun 2024

PONE-D-24-19911Research-informed decision-making for empowering integrated care system development: Co-creating innovative solutions to facilitate enhanced service provisionPLOS ONE

Dear Dr. Demir,

Thank you for submitting your manuscript to PLOS ONE. After careful consideration, we feel that it has merit but does not fully meet PLOS ONE’s publication criteria as it currently stands. Therefore, we invite you to submit a revised version of the manuscript that addresses the points raised during the review process. 

We look forward to receiving your revised manuscript.

Kind regards,

Reindolf Anokye

Academic Editor

PLOS ONE

Journal Requirements:

Reviewers' comments:

Reviewer's Responses to Questions

**Comments to the Author**

1. Is the manuscript technically sound, and do the data support the conclusions?

Reviewer #1: Yes

Reviewer #2: Partly

2. Has the statistical analysis been performed appropriately and rigorously?

Reviewer #1: Yes

Reviewer #2: I Don't Know

3. Have the authors made all data underlying the findings in their manuscript fully available?

Reviewer #1: No

Reviewer #2: Yes

4. Is the manuscript presented in an intelligible fashion and written in standard English?

Reviewer #1: No

Reviewer #2: Yes

5. Review Comments to the Author

**Reviewer #1:**  I would like to thank the authors for the possibility to read the paper.

I think that the topic is very interesting and that could become very appreciate both by the academic community and the practitioners.

Considering the valuable topics, I have some comments for the authors to improve their study.

• Introduction.

The introduction is very long and rich of topics, insights, literature references and comments. I think that this could derive from the objective of the study and could cause difficulties during the papers’ reading. For this reason, I could that could be better to provide an overview of the issue, considering the background and providing relevant examples, able also to describe and highlight the literature gaps or the possible practical contributions of the study, and then defining a paragraph of literature review for the collection of the already published evidence related to the Integrated Care Management model, enlarging the already defined paragraph about ICM in the UK, not presenting only the Policy and the Practice but presenting also the literature evidence about the topic.

In addition, in the Introduction, the systems approach is presented and described but this part of the manuscript is not well linked to the first part of the introduction, providing some issues for the reader, in terms of knowledge and contents’ flow.

Also, the concept of decision support tool (DST) could be better introduced and presented, if relevant from the beginning.

Moreover, the Table 1, presenting some questions on how OR could achieve enhanced ICM in a hospital setting, is not so clear. Are these the research questions of the papers? Or these are some guideline questions?

• Literature Review.

In this paragraph, I think that a summary Table could help the reader to focus on the different proposed alternatives.

• Materials and Methods.

- More references may be introduced to present and justify the methodological approach and research rigour.

- In the section of System Modelling, more details may be provided with reference to the conducted meetings and the focus group.

- The section devoted to Data Collection and Analysis is general and it is difficult to understand which dataset and information are used for the study as well as how the authors collected the data.

• Case Study: Application of the Model to an NHS Hospital.

The data included in the Table 2 might be moved into the Materials or could be better to clarify that this is the materials and methods section dedicated to the case, and then presenting the results.

It is not clear for my point of view, how the authors collected the data and which is the data source(s).

• Results

I think that it could be better to move the scenarios’ definition and description into the methodological section.

• Discussion

This section may be devoted to discuss the achieved results, and the methodological approach applied, in comparison with already published papers and reference, in order to explain the position.

In addition, the practical implications and the tool’s advantages for the KOL are well presented and structured, while the theoretical implications are lacking and might be presented.

Good luck for the study and for the publication!!

**Reviewer #2: ** Thank you, I enjoyed reading your manuscript. I think it is publishable, subjective to some corrections.

The study provides a valuable contribution to the field of integrated care system in healthcare delivery. However, the manuscript could be improved by enhancing the critical analysis in the literature review and discussion sections, and by ensuring conciseness and clarity throughout. In the Word document, I highlighted the areas that might benefits of some improvements.

- The introduction was well written and effectively sets the context by discussing global healthcare challenges and the importance of integrated care. However, it could be more concise. Some sections, like the NHS's history, could be shortened to maintain reader engagement.

- The literature review nicely links the theoretical framework to practical applications in healthcare, providing relevant examples and citations, but it could benefit from a more critical analysis and a better synthesis of the literature.

- Although the results section provides a thorough analysis of the impact of interventions on activity, resource utilisation, costs, and revenues, you could discuss the implications of the results here, linking back to yourstudy's objectives and pertinent literature.

- The discussion part highlights the practical impact of the DST on decision-making and service provision in the NHS. However, a more critical perspective is needed to address potential limitations and challenges in implementing the proposed interventions. It would also be useful to discuss the generalisability of the findings to other healthcare settings or countries with similar healthcare systems.

Well done again and fingers crossed you will have it publish soon.

6. PLOS authors have the option to publish the peer review history of their article (what does this mean? ). If published, this will include your full peer review and any attached files.

**Do you want your identity to be public for this peer review?** For information about this choice, including consent withdrawal, please see our Privacy Policy .

Reviewer #1: No

Reviewer #2: No

---

## [Author Response · Author response to Decision Letter 1]

17 Jul 2024

All responses to reviewers can be found in the uploaded file "Response to Reviewers".

---

## [Decision Letter · Decision Letter 1]

21 Nov 2024

PONE-D-24-19911R1Research-informed decision-making for empowering integrated care system development: Co-creating innovative solutions to facilitate enhanced service provisionPLOS ONE

Dear Dr. Demir,

Thank you for submitting your manuscript to PLOS ONE. After careful consideration, we feel that it has merit but does not fully meet PLOS ONE’s publication criteria as it currently stands. Therefore, we invite you to submit a revised version of the manuscript that addresses the points raised during the review process.

**In addition to the specific comments, please pay attention to the general comments by reviewers below**

Reviewer 1

The paragraph "ICM in the UK: Policy and Practice" is well conceived, but I think that this paragraph is not so connected with the other introductive sections, and this should be a problem for the paper's readiness. Why do you not think to introduce the concepts and the topics of this paragraph in the first part of the Introduction and then maintain the section of the Literature review?

Table 1 allows one to clarify better the role of the proposed questions in light of the background provided but anyway, I think that also one or two more precise research question(s) for the paper, connected with the objective defined by the authors, should be proposed to help the reader understanding all the paper

Considering that the authors stated in the methodological section that semi-structured interviews and focus groups were conducted, I think that some details regarding the questions posed or the information/areas investigated should help.

Another important point that I think is missing also in this updated version of the paper is related to the comparison of the presented study with other already published evidence. How does the paper differ from or confirm previous evidence?

Reviewer 2

The manuscript still lacks detailed information on the extent and nature of stakeholder involvement in the co-creation process with hospital staff. Providing more specifics about stakeholder engagement, the feedback process, and how their input influenced the development of the decision support tool (DST) would enhance the study's validity and relevance. Furthermore, the manuscript would benefit from a more thorough discussion of its limitations, including challenges related to data collection and potential biases in stakeholder engagement. Addressing these limitations would provide a more balanced perspective on the study's contributions and scope.

We look forward to receiving your revised manuscript.

Kind regards,

Moses Mukuru, Ph.D.

Academic Editor

PLOS ONE

Journal Requirements:

Reviewers' comments:

Reviewer's Responses to Questions

**Comments to the Author**

1. If the authors have adequately addressed your comments raised in a previous round of review and you feel that this manuscript is now acceptable for publication, you may indicate that here to bypass the “Comments to the Author” section, enter your conflict of interest statement in the “Confidential to Editor” section, and submit your "Accept" recommendation.

Reviewer #1: All comments have been addressed

Reviewer #2: All comments have been addressed

Reviewer #3: All comments have been addressed

2. Is the manuscript technically sound, and do the data support the conclusions?

Reviewer #1: Yes

Reviewer #2: Yes

Reviewer #3: Partly

3. Has the statistical analysis been performed appropriately and rigorously?

Reviewer #1: N/A

Reviewer #2: Yes

Reviewer #3: I Don't Know

4. Have the authors made all data underlying the findings in their manuscript fully available?

Reviewer #1: Yes

Reviewer #2: Yes

Reviewer #3: Yes

5. Is the manuscript presented in an intelligible fashion and written in standard English?

Reviewer #1: Yes

Reviewer #2: Yes

Reviewer #3: Yes

6. Review Comments to the Author

Reviewer #1: Dear Authors,

thank you very much for the possibility to read again your paper after the first round of revision.

I really appreciate the efforts that you adopted to improve your work.

I have some comments to finalize the work:

- the paragraph "ICM in the UK: Policy and Practice" is well conceived, but I think that this paragraph is not so connected with the other introductive sections, and this should be a problem for the paper's readiness. Why do you not think to introduce the concepts and the topics of this paragraph in the first part of the Introduction and then maintain the section of the Literature review?

- the Table 1 allows to better clarify the role of the proposed questions on the light of the background provided but anyway, I think that also one or two more precise research question(s) for the paper, connected with the objective defined by the authors, should be proposed to help the reader understanding all the paper

- considering that the authors stated in the methodological section that semi-structured interviews and focus group were conducted, I think that some details regarding the questions posed or the information/areas investigated should help.

- another important point that I think is missing also in this updated version of the paper is related to the comparison of the presented study with other already published evidence. How the paper differs from or confirm previous evidence?

Good luck for the publication!

Reviewer #2: Thank you for the opportunity to re-review this manuscript. I enjoyed reading it.

While the authors have addressed most of the previously raised comments, the manuscript still lacks detailed information on the extent and nature of stakeholder involvement in the co-creation process with hospital staff. Providing more specifics about stakeholder engagement, the feedback process, and how their input influenced the development of the decision support tool (DST) would enhance the study's validity and relevance. Furthermore, the manuscript would benefit from a more thorough discussion of its limitations, including challenges related to data collection and potential biases in stakeholder engagement. Addressing these limitations would provide a more balanced perspective on the study's contributions and scope.

Reviewer #3: .

7. PLOS authors have the option to publish the peer review history of their article (what does this mean? ). If published, this will include your full peer review and any attached files.

**Do you want your identity to be public for this peer review?** For information about this choice, including consent withdrawal, please see our Privacy Policy .

Reviewer #1: No

Reviewer #2: No

Reviewer #3: No

---

## [Author Response · Author response to Decision Letter 2]

23 Nov 2024

See the attached "Response to Reviewer" file for details.

---

## [Decision Letter · Decision Letter 2]

9 Mar 2025

PONE-D-24-19911R2Research-informed decision-making for empowering integrated care system development: Co-creating innovative solutions to facilitate enhanced service provisionPLOS ONE

Dear Dr. Demir,

Thank you for submitting your manuscript to PLOS ONE. After careful consideration, we feel that it has merit but does not fully meet PLOS ONE’s publication criteria as it currently stands. Therefore, we invite you to submit a revised version of the manuscript that addresses the points raised during the review process.

We look forward to receiving your revised manuscript.

Kind regards,

Dr. Mohammed Misbah Ul Haq, Pharm-D

Academic Editor

PLOS ONE

Reviewers' comments:

Reviewer's Responses to Questions

**Comments to the Author**

1. If the authors have adequately addressed your comments raised in a previous round of review and you feel that this manuscript is now acceptable for publication, you may indicate that here to bypass the “Comments to the Author” section, enter your conflict of interest statement in the “Confidential to Editor” section, and submit your "Accept" recommendation.

Reviewer #1: All comments have been addressed

Reviewer #2: (No Response)

2. Is the manuscript technically sound, and do the data support the conclusions?

Reviewer #1: Yes

Reviewer #2: No

3. Has the statistical analysis been performed appropriately and rigorously?

Reviewer #1: N/A

Reviewer #2: N/A

4. Have the authors made all data underlying the findings in their manuscript fully available?

Reviewer #1: Yes

Reviewer #2: Yes

5. Is the manuscript presented in an intelligible fashion and written in standard English?

Reviewer #1: Yes

Reviewer #2: Yes

6. Review Comments to the Author

Reviewer #1: (No Response)

Reviewer #2: Thank you for the revision. While the manuscript addresses an important topic, key concerns remain unresolved. The study heavily relies on literature review without sufficiently synthesizing key theoretical contributions, and a clear theoretical framework is lacking. Additionally, the methodology section lacks critical details, including the number of interviews and focus groups conducted, participant selection criteria, and the thematic analysis process for qualitative data. The absence of clarity on the sampling method, number of participants, and consideration of data saturation affects the study’s transparency and reproducibility.

The ethics statement mentions that "no ethics approval was required as the study did not involve human subjects," yet interviews and focus groups were conducted. Given these limitations, I regret to recommend rejection at this stage. However, I encourage the authors to refine the manuscript, enhance methodological rigor, and seek publication in a more suitable venue.

7. PLOS authors have the option to publish the peer review history of their article (what does this mean? ). If published, this will include your full peer review and any attached files.

**Do you want your identity to be public for this peer review?** For information about this choice, including consent withdrawal, please see our Privacy Policy .

Reviewer #1: No

Reviewer #2: No

---

## [Author Response · Author response to Decision Letter 3]

10 Mar 2025

See the response to reviewers file.

---

## [Editor Report · Decision Letter 3]

16 Mar 2025

Research-informed decision-making for empowering integrated care system development: Co-creating innovative solutions to facilitate enhanced service provision

PONE-D-24-19911R3

Dear Dr. Eren Demir,

We’re pleased to inform you that your manuscript has been judged scientifically suitable for publication and will be formally accepted for publication once it meets all outstanding technical requirements.

Kind regards,

Dr. Mohammed Misbah Ul Haq, Pharm-D

Academic Editor

PLOS ONE
---

## [Editor Report · Acceptance letter]

PONE-D-24-19911R3

PLOS ONE

Dear Dr. Demir,

I'm pleased to inform you that your manuscript has been deemed suitable for publication in PLOS ONE. Congratulations! Your manuscript is now being handed over to our production team.

Kind regards,

on behalf of

Dr. Mohammed Misbah Ul Haq

Academic Editor

PLOS ONE